# Immune Microenvironment in Sporadic Early-Onset versus Average-Onset Colorectal Cancer

**DOI:** 10.3390/cancers15051457

**Published:** 2023-02-24

**Authors:** Fanny Andric, Ala Al-Fairouzi, Yvonne Wettergren, Louis Szeponik, Elinor Bexe-Lindskog, James C. Cusack, Gerald Tumusiime, Marianne Quiding-Järbrink, David Ljungman

**Affiliations:** 1Department of Microbiology and Immunology, Institute of Biomedicine, Sahlgrenska Academy, University of Gothenburg, 412 96 Gothenburg, Sweden; 2Department of Surgery, Institute of Clinical Sciences, Sahlgrenska Academy, University of Gothenburg, 412 96 Gothenburg, Sweden; 3Department of Surgery, Sahlgrenska University Hospital, Region Västra Götaland, 413 45 Gothenburg, Sweden; 4Department of Surgery, Massachusetts General Hospital, Boston, MA 02114, USA; 5Department of Surgery, Uganda Christian University School of Medicine, Mukono P.O. Box 4, Uganda

**Keywords:** early-onset colorectal cancer, tumor microenvironment, tumor-infiltrating lymphocytes, gene expression, immunofluorescence

## Abstract

**Simple Summary:**

The incidence of non-hereditary cancer in the left colon and rectum is increasing in young patients worldwide for unknown reasons. To understand this phenomenon, the biology of early-onset colorectal cancer needs to be established. Here, we investigated the immune response to tumors by selecting a highly representative group of patients younger than 45 years matched to those aged 70–75 years, excluding hereditary cases. Both T-cell distribution in tumors and expression of 770 immune-related genes were overall similar between the groups. The findings suggest that the immune response in cancer of the left colon and rectum is not dependent on age and that early-onset colorectal cancer is likely not related to immune response deficiencies.

**Abstract:**

The incidence of left-sided colon and rectal cancer in young people are increasing worldwide, but its causes are poorly understood. It is not clear if the tumor microenvironment is dependent on age of onset, and little is known about the composition of tumor-infiltrating T cells in early-onset colorectal cancer (EOCRC). To address this, we investigated T-cell subsets and performed gene expression immune profiling in sporadic EOCRC tumors and matched average-onset colorectal cancer (AOCRC) tumors. Left-sided colon and rectal tumors from 40 cases were analyzed; 20 EOCRC (<45 years) patients were matched 1:1 to AOCRC (70–75 years) patients by gender, tumor location, and stage. Cases with germline pathogenic variants, inflammatory bowel disease or neoadjuvant-treated tumors were excluded. For T cells in tumors and stroma, a multiplex immunofluorescence assay combined with digital image analysis and machine learning algorithms was used. Immunological mediators in the tumor microenvironment were assessed by NanoString gene expression profiling of mRNA. Immunofluorescence revealed no significant difference between EOCRC and AOCRC with regard to infiltration of total T cells, conventional CD4^+^ and CD8^+^ T cells, regulatory T cells, or γδ T cells. Most T cells were located in the stroma in both EOCRC and AOCRC. Immune profiling by gene expression revealed higher expression in AOCRC of the immunoregulatory cytokine IL-10, the inhibitory NK cell receptors KIR3DL3 and KLRB1 (CD161), and IFN-a7 (IFNA7). In contrast, the interferon-induced gene IFIT2 was more highly expressed in EOCRC. However, in a global analysis of 770 tumor immunity genes, no significant differences could be detected. T-cell infiltration and expression of inflammatory mediators are similar in EOCRC and AOCRC. This may indicate that the immune response to cancer in left colon and rectum is not related to age of onset and that EOCRC is likely not driven by immune response deficiency.

## 1. Introduction

Colorectal cancer is the third most common cancer and the second cause of cancer mortality globally [1]. A disturbing increase in incidence before 50 years of age can be seen globally [2,3]. This early-onset colorectal cancer (EOCRC) currently constitutes about 15% of the total colorectal cancer burden but is expected and increase by 140% by 2040 to become the leading cause of cancer-related death among those under 50 years of age in the US [4,5].

Even though about a fifth of EOCRC cases are due to pathogenic germline variants, the epidemiologic surge is primarily constituted by sporadic tumors in the left colon and rectum [6]. Furthermore, the majority of EOCRC cases do not carry other well-known risk factors such as inflammatory bowel disease or a family history of colorectal malignancy [6]. Indeed, it has been speculated that EOCRC constitutes a distinct disease with features different from those of average-onset CRC (AOCRC). The full understanding needed for management of the EOCRC warrants investigation into the biologic properties of these tumors.

The immune response has been shown to be a part of the complex etiology of colorectal cancer. Generally, increased T-cell infiltration in tumor tissue and surrounding stroma of established tumors is associated with better overall survival [7]. Patients with a higher density of cytotoxic CD8^+^ T cells have been shown to relapse less than those with a low density. Further, the presence of memory T cells has been shown to correlate with an improved prognosis [7,8]. Taken together, the immune contexture (type, differentiation, density, and location of adaptive immune cells within distinct tumor regions) has been shown to be a stronger predictor of survival than the traditional TNM classification [7,9]. Hence, immune cell profiling, such as Immunoscore, is increasingly utilized to predict patient outcome.

To establish the mechanisms of EOCRC progression, it is necessary to further explore the immune response to these tumors. Some previous studies have suggested a lower lymphocyte infiltration into tumors from EOCRC patients and pathological features associated with a more aggressive growth pattern [8,10].

We believe that the interpretation of molecular and cellular differences present in available published cohorts may sometimes be hampered by a heterogenous case-mix, including cases with hereditary syndromes, IBD, and right-sided tumors. By focusing on a homogenous cohort representing the new epidemiologic surge, we hope to shed light on the biology of these EOCRC tumors.

In this study, we aimed to determine if there are significant differences between sporadic EOCRC and AOCRC with regard to the local tumor immune response. We thus investigated the presence and distribution of relevant subsets of T cells (total T-cell infiltration, cytotoxic CD8^+^ T cells, CD4^+^ T helper cells, gd T cells, regulatory T cells, and Granzyme B expression), along with the expression of a large set of genes of relevance to tumor progression and the immune microenvironment.

## 2. Materials and Methods

### 2.1. Study Design and Population

A database and biorepository of consecutive patients at the Surgical Oncology Laboratory at Sahlgrenska University Hospital were used to identify 20 EOCRC and 20 AOCRC cases between 2011 and 2020. A hierarchical cluster analysis was used to match the 20 early-onset cases to the 20 average-onset cases 1:1 for gender, tumor location, and stage, which were absolute inclusion criteria. Only EOCRC patients with left-sided colon and rectal tumors were included in the study, and we excluded patients with germline mutation syndromes for the purpose of focusing on the cases that constitute the increasing global incidence. Furthermore, no cases of inflammatory bowel disease were included. Acknowledging a likely biologic age continuum, an age selection of 18–45 years, representing distinct early-onset, and 70–75 years, representing the peak of traditional incidence, was made to achieve good discriminatory power between the groups. Tumors treated with neoadjuvant chemotherapy or radiotherapy were excluded in order to analyze treatment-naïve tissue only.

### 2.2. DNA and RNA Isolation

Tumor biopsies were snap-frozen in liquid nitrogen after removal and stored at −80 °C until used. Genomic DNA was isolated from tissues using the QIAamp DNA mini kit (Qiagen AB, Kista, Sweden, Cat. No. 51304) according to the manufacturer’s instructions. Total RNA was isolated from formalin-fixed paraffin-embedded (FFPE) tumor tissue using the RNeasy FFPE Kit (Qiagen AB, Kista, Sweden, Cat. No. 73504) according to the manufacturer’s instructions. The samples were kept at −20 °C until analysis.

### 2.3. MSI Analysis

The microsatellite status was analyzed using the MSI Analysis System, Version 1.2 (Promega, Nacka, Sweden), which includes fluorescently labeled primers for co-amplification of seven markers, including five mononucleotide repeat markers (BAT-25, BAT-26, NR-21, NR-24, and MONO-27) and two pentanucleotide repeat markers (Penta C and Penta D). Two ng of DNA was used in a 10 μL reaction volume that contained a fluorophore-labeled primer pair, Taq DNA polymerase, deoxyribonucleotide triphosphate mix, and buffer. PCR conditions were as follows: denaturation at 95 °C for 11 min, then at 96 °C for 1 min, followed by 10 cycles of 94 °C for 30 s, 58 °C for 30 s, and 70 °C for 1 min, and 20 cycles at 90 °C for 30 s, 58 °C for 30 s, and 70 °C for 1 min, then 60 °C for 30 min followed by a 4 °C soak. After amplification, 9.5 μL deionized formamide were combined with 0.5 μL of Internal Lane Standard 600 and 1 μL of the PCR reaction. This mixture was denatured at 95 °C for 3 min, chilled on ice, and spun briefly in a microcentrifuge. The microsatellite markers were detected on the ABI PRISM 3730 using PowerPlex 4C Matrix Standard (Cat. # DG4800). Microsatellite instability was defined as peak alterations in the marker electropherogram in the tumor compared with corresponding normal tissue. A tumor was defined as having MSI-H if more than one of the five markers showed instability, and as having MSI-L if only one marker showed instability. If no MSI was detected, the tumor was designated as MSS. Analysis of the MSI data was performed using Microsatellite Analysis Software version 1.1 (Thermo Fisher, Waltham, MA, USA).

### 2.4. Mutation Analysis

The KRAS G12D (GeneGlobe Cat. No. DMH0000286) and BRAF V600E (GeneGlobe Cat. No. DMH0000004) mutations were analyzed by digital PCR (dPCR) on a QIACuity-One 5-plex System instrument (Qiagen AB, Kista, Sweden). Template DNA (20 ng/reaction) was mixed in QIACuity Nanoplates (26K, Cat. No. 250001) with 10 µL of 4× QIACuity Probe PCR Master Mix (Cat. No. 250102), 1.3 µL of 30× dPCR LNA Mutation Assay (FAM/HEX, Cat. No. 250200), 1 µL of Hae III restriction enzyme (0.025 U/µL, TaKaRa BioEurope, France), and RNase-free water to a final volume of 40 µL. The plates were left at room temperature for 10 min before thermal cycling. The dPCR was run on the QIACuity instrument using the following cycling conditions: PCR initial heat activation for 2 min at 95 °C, followed by 40 cycles of denaturation for 15 s at 95 °C, and annealing/extension for 30 s at 60 °C. Data analyses were performed using the QIACuity Software Suite version 2.0, 2020.

### 2.5. Methylation Analysis

Two hundred ng of genomic DNA from each sample were used for bisulfite conversion using the EpiTecht Fast Bisulfite Kit (Qiagen, Sollentuna, Sweden) according to the manufacturer’s protocol. Gene-specific methylation of *MLH1*, *MGMT*, and *p16INK4a* was quantified using pyrosequencing and PyroMark Q24 assays (Qiagen, Sollentuna, Sweden). The regions to analyze were the following: *MLH1*, −209 to −181 from the transcription start site; *MGMT*, +17 to +39 in exon 1; and *p16INK4a*, +68 to +120 in exon 1. Global methylation was quantified using the PyroMark Q24 CpG LINE-1 assay (Qiagen, Sollentuna, Sweden). Information about the assays, including sequence to analyze, number of CpG sites and PCR conditions, are presented in Appendix A.

One μL of Sepharose beads were mixed with 40 μL of binding buffer and 22 μL of water in an Eppendorf tube. Sixty μL of this mix was added to 20 μL of PCR products in a 96-well plate and agitated at 1500 rpm for 10 min. The PyroMark Advanced Q24 Plate was filled with 0.375 μM of sequencing primer in 20 μL of annealing buffer. The washes were performed using the vacuum station according to the manufacturer’s instructions. To anneal the samples to sequencing primers, the temperature was increased to 80 °C for 5 min (*MLH1*) or 2.5 min (*MGMT*, *p16INK4a*, and LINE-1). The samples were then immediately processed in the PyroMark Advanced Q24 instrument.

Pyrosequencing of the purified single-stranded PCR products and CpG site quantification were accomplished using the PyroMark Q24 and related software (Qiagen). Each CpG site was assigned a percentage of methylation by evaluating the C/T ratio. The mean percentage of methylation across the CpG sites was calculated for each sample and each analyzed DNA sequence.

### 2.6. Immunofluorescence

Formalin-fixed paraffin-embedded slides were deparaffinized with xylene for 10 min and rehydrated in ethanol. Antigen retrieval was achieved with antigen retrieval (AR) buffer (10 mM Tris, 1 mM EDTA, 0,05% Tween 20, pH 9.0) in a 2100 Antigen Retriever (Aptum Biologics, Southampton, UK), where the slides were heated to 121 °C for 30 min, followed by a wash in PBS. Tissue sections were incubated with primary antibodies for 60 min at room temperature (RT), washed in PBS for 6 min, and incubated with Akoya anti-mouse/ anti-rabbit polymer for 10 min at RT, followed by incubation with Tyramide-fluorochrome conjugates for 10 min at RT, using the Opal-7 color Manual IHC Kit (Akoya Bioscience, Marlborough, MA, USA) according to the manufacturer’s instructions. Antibody stripping was performed in the AR buffer, and sections were microwaved at high power until boiling point was reached, followed by microwaving for an additional 10 min at 75 degrees. These steps were performed for each primary antibody in the following order: FoxP3, GrzB, TCRγδ, PanCK, CD8, and CD3. Antibody details and dilutions are available in Appendix A. Subsequently, nuclei were stained with DAPI (Invitrogen, Waltham, MA, USA) for 15 min at RT, and slides were mounted with prolonged artifice glass medium (Thermo Fisher Waltham, MA, USA) and #1,5 color glasses and sealed with nail polish. Tissue sections were scanned with TissueFAX CHROMA (TissueGnostics, Vienna, Austria) using an Axio Imager, a Z2 Microscope, a 20×/0.8/air objective (Zeiss, Oberkochen, Germany), and SpectraSplit filter set (Kromnigon, Mölndal, Sweden). The areas in the whole tumor scans, focused on the tumor center and not the invasive margin, were defined as tumor or stroma, and different cell subsets in the respective areas were quantified by Strataquest (TissueGnostics, Vienna, Austria) and expressed as cell number per mm^2^.

### 2.7. Gene Expression Analysis

The NanoString nCounter PanCancer Immune Profiling Panel (NanoString Technologies, Seattle, WA, USA) was used to assess the expression of 770 genes. Each biotinylated capture probe in the panel was manufactured with specificity for a 100-base region of the target mRNA. A reporter probe tagged with a fluorescent barcode was also included, thus resulting in two sequence-specific probes for each target transcript. Probes were hybridized to 60 ng of total RNA for 20 h at 65 °C and applied to the nCounter preparation station for automated removal of excess probe and immobilization of probe-transcript complexes on a streptavidin-coated cartridge. Data were collected with the nCounter digital analyzer by counting individual barcodes and analyzed using the software nSolver version 4.0.

### 2.8. Statistical Analyses

Statistical analyses were performed in GraphPad Prism, JMP, SPSS, and STATA. The chi-squared test or Fisher’s exact test was used as appropriate to compare categorical data between age groups; the Mann–Whitney U or Kruskal–Wallis test was used as appropriate to compare continuous data; and the log-rank test was used for Kaplan–Meier analysis. The *p* values of <0.05 were considered significant, and in nCounter gene expression analysis, Benjamini–Yekutieli false discovery rate (FDR) correction was utilized.

## 3. Results

### 3.1. Clinical, Pathological and Molecular Characteristics

From 779 well-characterized left-sided colorectal cancer cases, 20 EOCRC were identified by application of strict inclusion and exclusion criteria to represent the increasing incidence and a control group of 20 matched representative AOCRC. The clinical, pathological, and molecular characteristics of the two groups are summarized in Table 1.

### 3.2. T-Cell Infiltration in Early and Average-Onset CRC

T cells were defined as CD3^+^ mononuclear cells and enumerated by immunofluorescence in tumor tissue from patients with EOCRC and AOCRC. In the analysis, we separated tumor tissue, i.e., actual collections of tumor cells detected with the epithelia marker PanCK, and surrounding stroma. The density of T cells varied considerably between individuals, but the initial analyses revealed no significant difference between the two patient groups with regard to total T-cell infiltration in neither tumor nor stroma (Figure 1A,B). The majority of T cells were located in the tumor stroma in both early-onset (median 79%, range 56% to 95%) and average-onset CRC (80%, range 49% to 96%), and there was no significant difference between the two groups in this regard (Figure 1C,D).

### 3.3. Distribution of CD8^+^ T Cells in Early and Average-Onset CRC Tumors

Previous studies show that increased infiltration of cytotoxic T cells correlates with better survival in CRC patients [7,9,11], and we therefore assessed whether the density of tumor-infiltrating conventional CD8^+^ T cells (defined as CD3^+^CD8^+^TCRγδ^−^ cells) varies between the two patient groups. Analysis of CD8^+^ T cells in the tumor and surrounding stroma showed similar cell densities between patients diagnosed with early and average-onset CRC in both locations (Figure 2A–C). As cytotoxicity mediated by CD8^+^ T cells is dependent on cell-cell contacts, we also determined the stromal CD8^+^ T cells’ mean distance to the nearest tumor cell. However, this distance was not significantly different for cytotoxic T cells in EOCRC compared to AOCRC (Figure 2D). The cytotoxic potential of CD8^+^ T cells was determined by the expression of Granzyme B (GrB; Figure 2E). Only a minority of CD8^+^ T cells were GrB^+^, and again, the proportion of GrB^+^ T cells among all CD8^+^ T cells located in both tumor and stroma did not differ between the patient groups (Figure 2F,G). Together, these findings indicate that the number of CD8^+^ T cells and their cytotoxic potential do not differ between the patient groups.

### 3.4. CD4^+^ T Helper Cell and Treg Infiltration in Early and Average-Onset CRC Tumors

To assess conventional T helper (Th) and Treg cells in the tumors, Treg were identified as TCRγδ^−^CD3^+^CD8^−^FoxP3^+^ T cells, while TCRγδ^−^CD3^+^CD8^−^FoxP3^−^ T cells were identified as Th cells. Control staining confirmed the absence of CD8^+^CD4^+^ double-positive T cells in the tissues, as well as very low numbers of CD4^−^CD8^−^ double-negative T cells (Appendix A), except for gammadelta (γδ) T cells. Thus, the use of CD3^+^CD8^−^ status as a proxy marker for CD4^+^ T cells is valid when γδ T cells are excluded from the analysis. Our analyses confirmed that Th cells are by far the most abundant T-cell population in both tumors and stroma in both patient groups (Figure 3A–C). Furthermore, the frequencies of Th cells in tumor and stroma were similar between patient groups.

In contrast to cytotoxic T cells, Treg presence and function usually correlate with poor prognosis in CRC [12,13,14,15]. In our patient cohort, Treg comprised about 4–8% of all T cells in both tumor and stroma, and there were similar densities of Treg present in both tumors and stroma from early and average-onset CRC (Figure 3D,E). Previous studies have suggested the presence of CD8^+^FoxP3^+^ T cells with suppressive ability in CRC [16]. However, we could not detect any CD8^+^FoxP3^+^ T cells, neither in the stroma nor in the tumor mass.

### 3.5. γδ T Cells in Early and Average-Onset CRC Tumors

γδ T cells may have a potential role in antitumor immunity, although some evidence suggests that they may also have tumor-promoting effects [17]. Still, the role of γδ T cells in colorectal cancer immunity is largely unknown. We therefore decided to further investigate these cells and the potential differences between early and average-onset CRC. When densities of γδ T cells infiltrating the intestinal tumor and surrounding stroma were determined, we found somewhat more gd T cells in the stroma than in the tumor masses, but the difference was not as pronounced as with conventional CD4^+^ and CD8^+^ T cells (Figure 4A–C). As with the other T cell populations, there were no significant differences in the density of γδ T cells in EOCRC and AOCRC (Figure 4A,B). The large majority of γδ T cells were CD8^−^, both in the tumor and the stroma (median 99% and 100%, respectively). Furthermore, the expression of GrB by γδ T cells was higher than in conventional CD8^+^ T cells but did not differ between early and late-onset CRC patients in neither tumor nor stromal tissue (Appendix A). In addition, the mean distance from tumor cells of the stromal γδ T cells did not differ significantly between the two patient groups (Appendix A). As the actual numbers of γδ T cells were sometimes very low, we only used sections with more than 50 γδ T cells in the respective compartment for these calculations.

### 3.6. Immune Profile in Early and Average-Onset CRC Tumors

To obtain a comprehensive overview of immune reactions and mediators relevant for tumor immunity, we interrogated the expression of 770 genes in RNA extracted from formalin-fixed tumor sections from early and average-onset CRC. Three gene panels implicated in immune regulatory functions were preselected: a Regulatory panel, a Cytotoxicity panel, and a Proinflammatory panel, respectively (list of genes in Appendix A). Using this approach, we found that the immunoregulatory gene cytokine IL-10 was more abundantly expressed in AOCRC tumors than EOCRC. Furthermore, the inhibitory NK cell receptor genes KIR3DL3 and KLRB1 (CD161) were also more highly expressed in AOCRC, as was the gene for IFN-a7 (IFNA7) (Figure 5). In contrast, the interferon-induced gene IFIT2 was more highly expressed in EOCRC.

When employing the nCounter PanCancer Immune Profiling Panel on the full set of 770 genes, the immunofluorescence results were confirmed in the sense that there was no difference in the expression levels of CD3, CD4, CD8, TCRgd, FoxP3, or GrB between EORCR and AOCRC tumors. In fact, after FDR correction, there were no significant differences in the expression levels of any of the genes analyzed (Figure 6).

### 3.7. Correlation between Immune Parameters and Patient Outcome

The relapse rate and cancer-associated deaths are lower in CRC than many other tumor types. This was evident also in the current patient cohort during a three-year follow-up period of the 36 patients with stage I–III tumors (Figure 7A). In our material, there was no significant difference between the patient groups with regard to relapse-free survival. It is notable, however, that younger patients received more aggressive treatment (Table 1). We also examined the potential influence of T-cell infiltration into tumors on patient outcome by comparing cell densities between patients who experienced a relapse and relapse-free patients presenting with EOCRC or AOCRC. In this relatively limited material, there were no significant differences when comparing the infiltration into the actual tumor mass of all CD3^+^ T cells, CD8^+^ cytotoxic T cells, Treg, or γδ T cells between relapsing and relapse-free patients (Figure 7B–E). However, a trend towards lower CD3^+^ and CD8^+^ T-cell infiltration into tumors was observed in the relapsing patients.

## 4. Discussion

In this study, we provide a detailed comparison of T-cell infiltration in tumor tissues of patients with EOCRC and AOCRC. Our analyses clearly show that there are no general differences in T-cell infiltration between early and average-onset CRC tumors in the analyzed key subtypes.

It is now generally accepted that lymphocyte infiltration into tumors is a powerful prognostic factor, both in MSI-H and MSS tumors, and likely more important than tumor stage or mutational pattern to predict patient outcome [7,18,19]. Lymphocyte infiltration is generally higher in MSI-H tumors, but a subpopulation of MSS tumors with dense lymphocyte infiltration has also been described, and these patients have a disease-specific survival similar to those with MSI-H tumors and high lymphocyte infiltration [18]. Much like previous studies, we found a large variation in T-cell infiltration, and there was also a tendency toward a better outcome for the patients with larger T-cell infiltration. When comparing EOCRC and AOCRC tumors, however, there were no differences in total T-cell frequencies or the T-cell distribution between tumor and stroma.

In addition to total T-cell infiltration, the presence of CD8^+^ cytotoxic T cells is a major beneficial prognostic marker in CRC [9,20]. Here, we determined both total CD8^+^ T-cell infiltration and the presence of GrB-expressing putative effector cells with cytotoxic ability. As cytotoxic cells need close cell-cell contact to kill tumor cells, we also measured the mean distance to the nearest tumor cells. These analyses all showed a similarity between EOCRC and AOCRC, leading to the conclusion that there are no major differences in the influx of potentially beneficial T cells into distal colorectal tumors diagnosed at different ages. Furthermore, the presence of γδ T cells, another subset of cytotoxic cells with an unclear significance for CRC progression [21], was also similar between patient groups.

Not all T cells are beneficial and Treg frequencies often correlate with a poor prognosis in solid tumors [22]. In CRC, the effect of tumor-infiltrating Treg is less clear, with studies showing both a positive and negative influence on patient outcome [13,14,15,17,23]. Nevertheless, the presence of Treg cells in tumors from early-onset and average-onset CRC patients was similar. However, our mRNA analyses showed somewhat lower transcription of the immunoregulatory cytokine IL-10 in EOCRC. This may indicate that even if similar numbers of Treg are present in EOCRC and AOCRC tumors, their activity may be higher in the tumors from the older patients, which in turn may impede ongoing T-cell responses.

Expression analyses indicated that there may be certain significant differences that are valuable to confirm in other cohorts. Our interpretation is however that there is an overall similar immune response in the two groups as measured by the expression of genes relevant for tumor immunology.

Taken together, our findings indicate there is no difference in the T-cell infiltration between EOCRC and AOCRC patients. This needs to be confirmed in a larger sampling, but these findings are in line with results from a recent study by Ugai et al. [8], who found lower infiltration of lymphocytes in routine hematoxylin-stained sections from tumors in young CRC patients, but when comparing the densities of different types of T cells, there were no significant differences between younger and older patients [8]. This study by Ugai et al. did not utilize strict selection of consecutive patients, and therefore the patient groups were not homogenous. In addition, the intraepithelial and stromal cell densities were combined, which may give less useful information, as CD8^+^ T-cell infiltration into tumor epithelium provides stronger prognostic information than the total CD8^+^ infiltration into the tumor, especially in MSS tumors [24]. There have been suggestions that early and late-onset CRC may be due to specific mutations or deletions [25] or other risk factors concentrated in the younger population [26]. However, our data clearly show that age at presentation does not influence the T-cell infiltration at the time of diagnosis.

Based on mRNA data, it has also been proposed that young patients with CRC, especially with rectal tumors, have a more pronounced innate-immune response with increased complement and acute-phase reactants [27]. In our study, we found no global differences in the broad expression of inflammation-related markers but somewhat reduced expression of genes for NK cell receptors KIR3DL3 and KLRD1, IFIT2 and IFN-a7, that may suggest a mildly impaired innate immune response in EOCRC.

This study has several strengths. A key factor for investigating the new epidemiology of EOCRC is the homogenous patient material with regard to tumor type, stage, mutations, and location that is well characterized. This is an important aspect, as CRC comprises a heterogenous group of tumors [9,28]. Furthermore, we have analyzed lymphocyte infiltration in distinct anatomical compartments, i.e., tumor epithelium and stroma, and added expression data for comprehensive assessment. However, a possible weakness is the low number of patients included. As there is a large inter-individual variation, there might be a risk that we missed subtle differences between the immune responses in the two patient groups. However, we believe the large variation and general overlap in the results between the groups make it likely that there is no clinically significant age-dependent difference. As we have selected a homogenous group of patients with AOCRC, we acknowledge that describing the immune landscape in other subgroups of colorectal cancer, such as the extremely old or patients with signet-ring cell cancers, is highly relevant for a general understanding of colorectal cancer biology and warrants further investigation [29,30].

## 5. Conclusions

In summary, our results indicate that there are no significant differences with regard to T-cell infiltration between sporadic early-onset or average-onset cancer of the left colon and rectum, neither in the stroma nor in the tumor mass. Furthermore, expression analysis of 770 immune-related genes suggests no significant difference between the groups. We therefore predict that the knowledge acquired for CRC in general about the immune response and the importance of different T-cell subsets in the progression or control of CRC may be applicable also to young patients presenting with sporadic colorectal tumors.

## Figures and Tables

**Figure 1 cancers-15-01457-f001:**
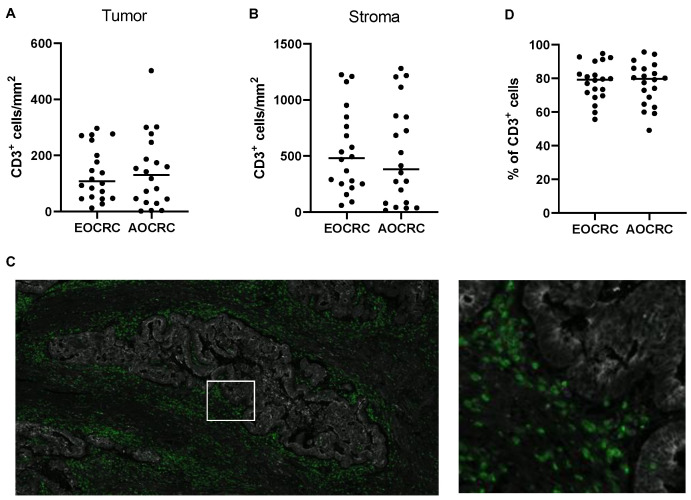
Density of CD3^+^ T cells. The density of CD3^+^ T cells in tumors from early-onset (EOCRC) and average-onset colorectal cancer (AOCRC) patients was determined by immunofluorescence in tumor tissue (**A**) and tumor stroma (**B**). (**C**) Representative immunofluorescence image from an EOCRC patient with tumor cells (panCK, grey) and CD3^+^ T cells (green). Magnification 50× left and 300× right. (**D**) Percentage of all CD3^+^ cells located in the stroma. Symbols show individual values, and lines indicate the median.

**Figure 2 cancers-15-01457-f002:**
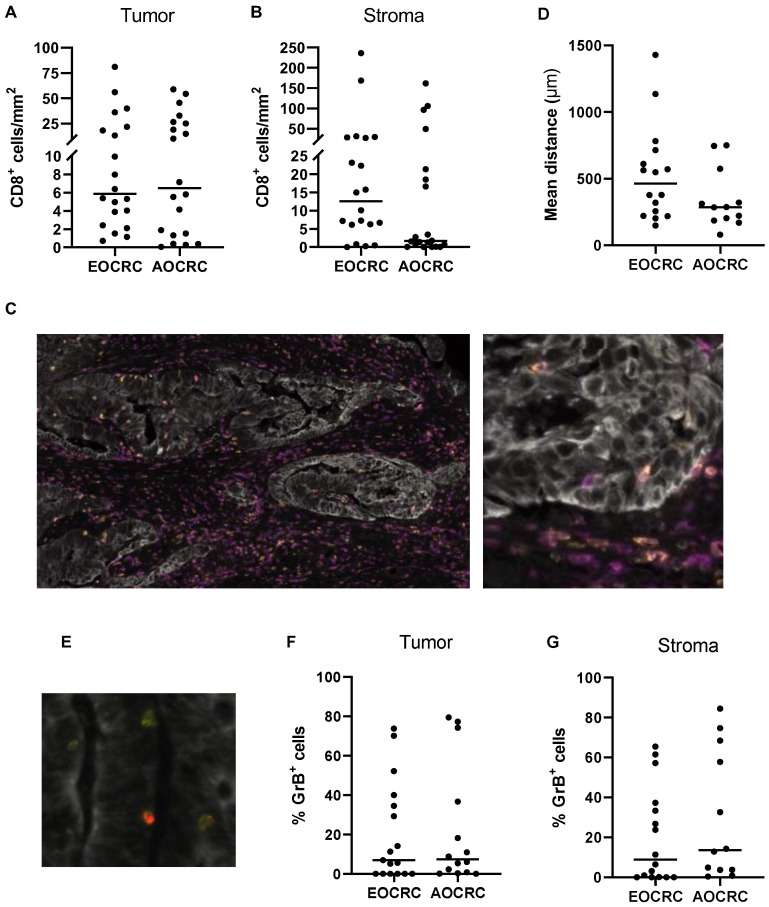
CD8^+^ T cells in the tumor microenvironment. The density of CD8^+^ T cells in tumors from early-onset (EOCRC) and average-onset colorectal cancer (AOCRC) patients was determined by immunofluorescence in tumor tissue (**A**) and tumor stroma (**B**). (**C**) Representative immunofluorescence image from an EOCRC patient with tumor cells (panCK, grey), CD3^+^ T cells (magenta), and CD8^+^ T cells (yellow). Magnification 100× (left) and 300× (right). (**D**) Mean distance of stromal CD8^+^ cells to the nearest tumor cell. (**E**) Representative immunofluorescence image with tumor cells (panCK, grey), CD8^+^ T cells (yellow), and GrB^+^ cells (red). Magnification 500×. Percentage of CD8^+^ cells that express Granzyme B (GrB) located in tumor tissue (**F**) and tumor stroma (**G**). Symbols show individual values, and lines indicate the median.

**Figure 3 cancers-15-01457-f003:**
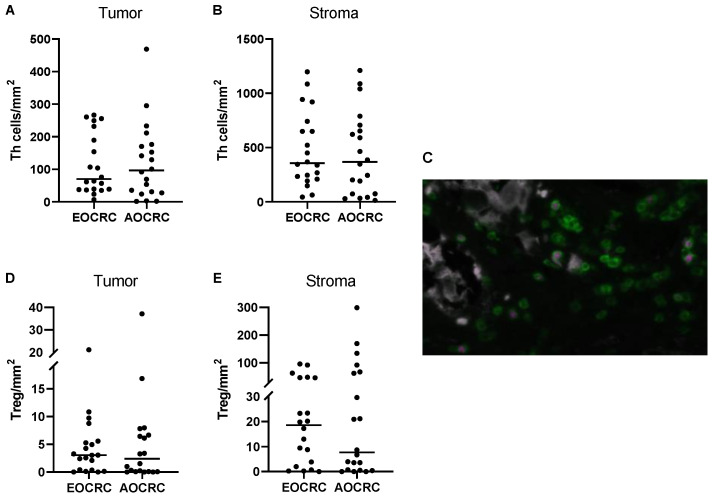
CD4^+^ T cells in the tumor microenvironment. The density of Th cells (CD3^+^CD8^−^FoxP3^−^ cells) and Treg (CD3^+^CD8^−^FoxP3^+^ cells) in tumors from early-onset (EOCRC) and average-onset colorectal cancer (AOCRC) patients was determined by immunofluorescence in tumor tissue (**A**,**D**) and tumor stroma (**B**,**E**). (**C**) Representative immunofluorescence image from an AOCRC patient with tumor cells (panCK, grey), CD3^+^ T cells (green), and FoxP3^+^ Treg (magenta). Magnification 300×. Symbols show individual values, and lines indicate the median.

**Figure 4 cancers-15-01457-f004:**
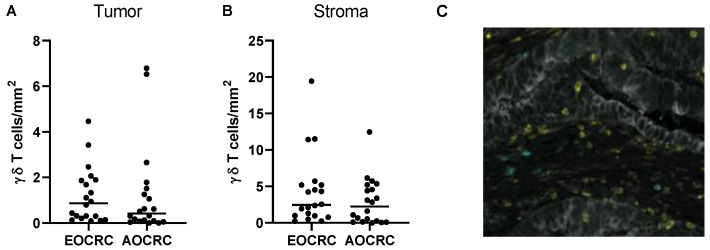
γδ T cells in the tumor microenvironment. The density of γδ T cells in tumors from early onset (EOCRC) and average-onset colorectal cancer (AOCRC) patients was determined by immunofluorescence in tumor tissue (**A**) and tumor stroma (**B**). (**C**) Representative immunofluorescence image from an EOCRC patient with tumor cells (panCK, grey), CD8^+^ T cells (yellow), and γδ T cells (turquoise). Magnification 260×. Symbols show individual values, and lines indicate the median.

**Figure 5 cancers-15-01457-f005:**
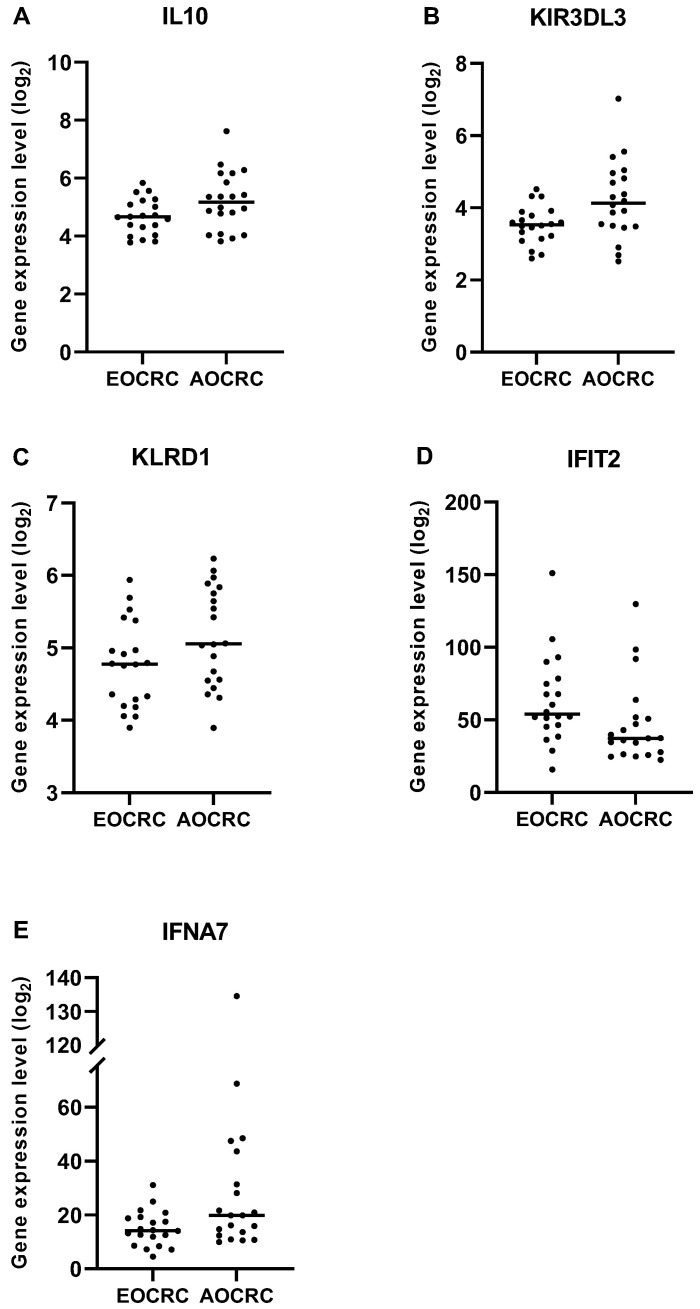
Gene expression in early and average-onset CRC. Significantly different expression in EOCRC and AOCRC in the three preselected gene panels are illustrated. Regulatory panel; (**A**) IL10 *p* = 0.0499, Cytotoxicity panel; (**B**) KIR3DL3 *p* = 0.0315 and (**C**) KLRD1 *p* = 0.0499, Pro-inflammatory panel; (**D**) IFIT2 *p* = 0.0167 and (**E**) IFNA7 *p* = 0.0439. Symbols show individual values, and lines indicate the median.

**Figure 6 cancers-15-01457-f006:**
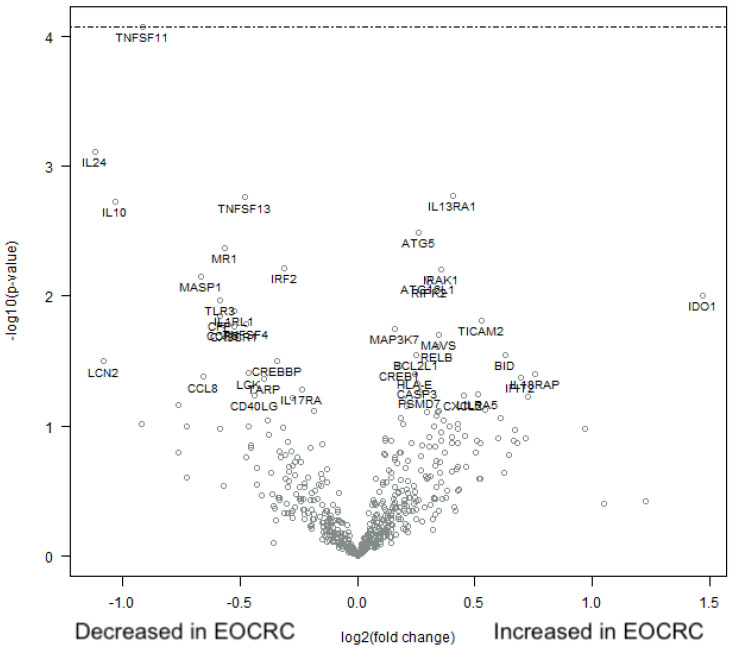
Immune profile in early and average-onset CRC tumors. RNA was extracted from formalin-fixed tumor tissue sections and analyzed using nCounter PanCancer Immune Profiling Panel. A volcano plot illustrating the relative up- and down-regulation of 770 genes in EOCRC tumors with AOCRC tumors as the baseline *n* = 20 in each group, respectively. The dashed line for the TNSF11 gene indicates *p* = 0.32 (32% false discoveries) in adjustment taking biologic gene-gene interactions into account.

**Figure 7 cancers-15-01457-f007:**
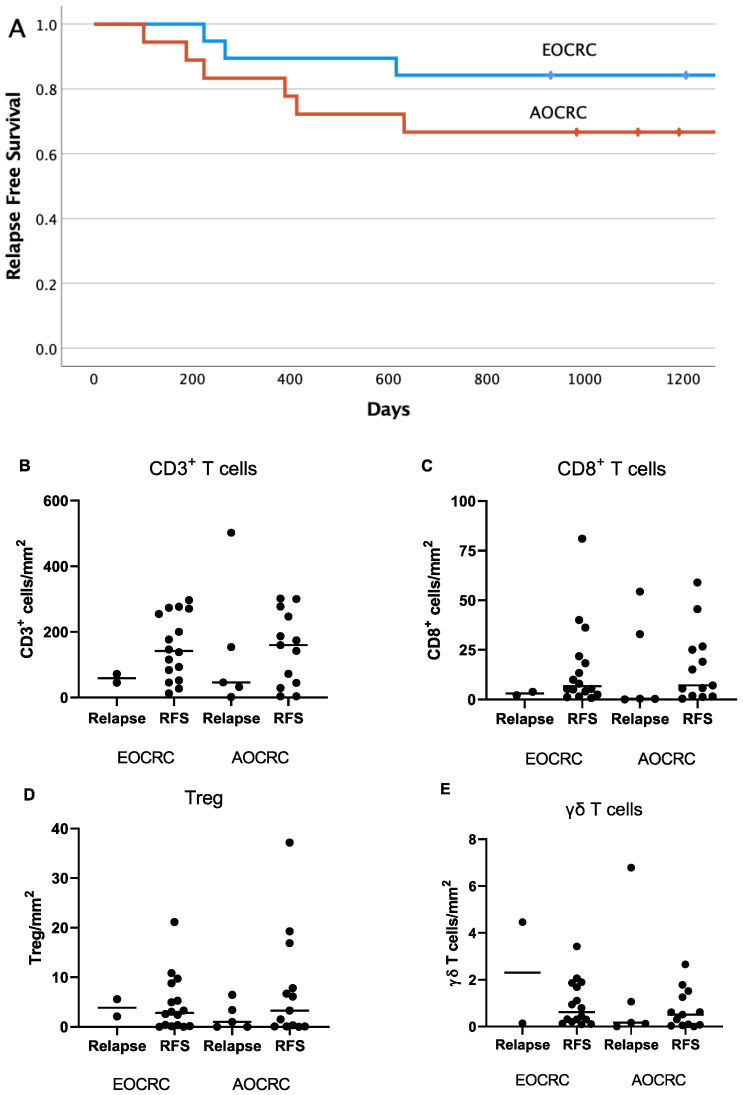
Intratumoral T-cell infiltration in relation to patient outcome. Kaplan–Meier plot (**A**) showing relapse-free survival in EOCRC (blue) and AOCRC (red) patients with stage I–III tumors during a 3-year follow-up period (*p* = 0.41 between groups). The density of all T cells (**B**), CD8^+^ T cells (**C**), Treg (**D**), and gd T cells (**E**) in tumors from EOCRC and AOCRC patients who experienced a relapse within 3 years or had a relapse-free survival (RFS) was determined by immunofluorescence. Symbols show individual values, and lines indicate the median.

**Table 1 cancers-15-01457-t001:** Clinical, pathological, and molecular characteristics of the groups according to age at diagnosis. EOCRC = 18–45 years, and AOCRC = 70–75 years.

Variable	EOCRC *n* = 20	AOCRC *n* = 20	*p* Value	Total *n* = 40
Sex *n* (%)				
Male	10 (50%)	10 (50%)	1.00	20 (50%)
Female	10 (50%)	10 (50%)		20 (50%)
Tumor localization *n* (%)				
Left-sided Colon	13 (65%)	13 (65%)	0.63	26 (65%)
Rectum	7 (35%)	7 (35%)		14 (35%)
Tumor differentiation (%)				
High grade	5 (25%)	1 (5%)	0.34	6 (15%)
Low grade	13 (65%)	17 (85%)		30 (76%)
Mucinous	2 (10%)	2 (10%)		4 (10%)
Tumor stage *n* (%)				
Stage I	2 (10%)	2 (10%)	1.00	4 (10%)
Stage II	7 (35%)	7 (35%)		14 (35%)
Stage III	9 (45%)	9 (45%)		18 (45%)
Stage IV	2 (10%)	2 (10%)		4 (10%)
Tumor				
pT1	0 (0%)	2 (10%)	0.38	2 (5%)
pT2	3 (15%)	4 (20%)		7 (18%)
pT3	16 (80%)	12 (60%)		28 (70%)
pT4	1 (5%)	2 (10%)		3 (8%)
Nodes				
pN0	10 (50%)	10 (50%)	1.00	20 (50%)
pN1	5 (25%)	5 (25%)		10 (25%)
pN2	5 (25%)	5 (25%)		10 (25%)
Number of examined lymph nodes; mean (SD)	27 (11)	24 (13)	0.04	25.5 (12)
Number of positive lymph nodes; mean (SD)	2 (3)	3 (4)	0.87	2 (3)
Lymph node ratio; mean % (SD)	6 (11)	10 (15)	0.76	8 (13)
dMMR **, *n*	1 (5%)	1 (5%)	1.00	2 (5%)
BRAF V600E mutation, *n*	5 (25%)	2 (10%)	0.20	7 (18%)
KRAS mutation, *n*	10 (50%)	9 (45%)	0.75	19 (48%)
MLH1 methylation, *n*	0 (0%)	1 (5%)	1.00	1 (3%)
MGMT methylation (SD)	16 (24)	16 (25)	0.89	16 (24)
P16INK4a methylation (SD)	4 (5)	13 (19)	0.97	9 (15)
LINE1 methylation (SD)	68 (8)	70 (7)	0.44	69(7)
Adjuvant chemotherapy				
5-FU	5 (25%)	7 (35%)	0.20	12 (30%)
5 + FU + Oxaliplatin	5 (25%)	1 (5%)		6 (15%)
No chemotherapy	10 (50%)	12 (60%)		22 (55%)
Relapse at three years *n*			0.56	
Yes	3 (16%)	5 (28%)		8 (22%)
No	16 (84%)	13 (72%)		29 (78%) *

* 3 patients in stage 4 were never declared tumor-free and therefore excluded from outcome analysis. ** dMMR was found in 1 AOCRC with MLH1 promotor methylation and 1 EOCRC without germline mutation or MLH1 methylation as a de novo mutation.

## Data Availability

Data can be accessed from authors upon request.

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
