# Peer review of "Immune Microenvironment in Sporadic Early-Onset versus Average-Onset Colorectal Cancer"

_cancers, 2023, doi:10.3390/cancers15051457_

Round 1

Reviewer 1 Report

The study is well done, the material is large enough and the methods look reliable. However the study is based on extensive and very recent literature, gives some new information and this warrants its publication.

Author Response

We thank the reviewer for the positive evaluation of our work.

Reviewer 2 Report

The manuscript by Andric et al. examines whether the population of immune cells in the tumor microenvironment may be different between early- (EOCRC) and late-(AOCRC) onset colorectal cancer.  Clear understanding of immune cell composition within the tumor stroma provides significant benefits both as a biomarker and as a potential target of chemoprevention.  The study is simple and straight-forward, and the manuscript is well-written.   Not only the quantitative data on T cell infiltration, but the results were also confirmed with gene expression analysis.  Although there were no differences in the T cell populations, ethiology of EOCRC is still unknown that the study provided good information for the field.

Author Response

(The authors gave the same response as above.)

Reviewer 3 Report

Andric et al. investigate the Immune microenvironment, including the changes of CD4 and CD8, cytokines and etc., in sporadic early-onset versus average-onset colorectal cancer. The manuscript is well-written but the methodology is too simple: for example, it has been well-known that the alterations of immune microenvironment are associated with the biologic phenotype characteristics of the tumor such as MSI, mucin-producing, ...; and not just dichomize and analyze by patient age and tumor sidedness. I would like to know the immune microenvironment in the extremely old patients (age ≥ 90 years) and the  signet ring cell carcinomas. I suggest the citation the following 2 articles and make some discussions in the direction of immune microenvironment:

1.Asian Journal of Surgery Volume 45, Issue 1, January 2022, Pages 208-212.

2.Asian Journal of Surgery Volume 44, Issue 1, January 2021, Pages 105-110.

Author Response

  1. Andric et al. investigate the Immune microenvironment, including the changes of CD4 and CD8, cytokines and etc., in sporadic early-onset versus average-onset colorectal cancer. The manuscript is well-written but the methodology is too simple: for example, it has been well-known that the alterations of immune microenvironment are associated with the biologic phenotype characteristics of the tumor such as MSI, mucin-producing...; and not just dichomize and analyze by patient age and tumor sidedness. I would like to know the immune microenvironment in the extremely old patients (age ≥ 90 years) and the  signet ring cell carcinomas. I suggest the citation the following 2 articles and make some discussions in the direction of immune microenvironment:

1.Asian Journal of Surgery Volume 45, Issue 1, January 2022, Pages 208-212. 

2.Asian Journal of Surgery Volume 44, Issue 1, January 2021, Pages 105-110. 

Dear reviewer, thank you for your pertinent comments about the multiple factors influencing the immune microenvironment. It was precisely this multifactorial situation that was our concern when we read some other investigations into early-onset colorectal cancer (EOCRC). We believe that, despite the best intentions, sometimes more complicated analysis plans may lead to confusion and false conclusions. Simplicity in design provides robustness and transparency. If for example the cohorts were expanded to be more inclusive and heterogenous they would have to be significantly larger and would carry an increased risk of not capturing all factors into the multivariate analysis.

In this study we were interested in explaining the recent global trend of the increasing EOCRC incidence. Therefore, we set out to make the study design as simple as possible in order to isolate the one factor of age-of-onset. We excluded hereditary cases, the heterogenous and atypical MSI-prone right sided colon cancers, cases with inflammatory bowel disease, neoadjuvantly treated rectal cancers and made sure the MSI status and several other molecular characteristics were known to be certain that we had an EOCRC cohort representative of the new epidemiology. We finally compared these to an AOCRC cohort matched for key well-established factors. We have now tried to explain this strategy better in the revised manuscript. 

We agree fully that describing the immune landscape in other subgroups of colorectal cancer, such as in the extremely old or in patients with signet ring cell cancers, is highly relevant and worth further investigation. To take these aspects into account and improve the manuscript we have revised the discussion section with references to the two articles you suggested.

Reviewer 4 Report

Authors compare early onset <45 years of age and average onset CRC. They focus specifically on immune cell infiltration, namely T cell populations and Nanostring N counter analysis. The hypothesis tested is an immune predisposition to disease. The authors fail to find any supporting evidence across 20 tumours in each group for this hypothesis. The numbers included are small. The tumours are matched, but formal propensity scoring does not appear to have been done? The findings are unsurprising. The groups are small and not powered to observe a difference. The immunoscore or equivalent is not presented. The site of infiltrate is not stated and this can influence outcome. Overall this paper in my opinion does not merit publication in its present form due to lack of sufficient interest, though I acknowledge the robust nature of the processes employed. 

Author Response

Authors compare early onset <45 years of age and average onset CRC. They focus specifically on immune cell infiltration, namely T cell populations and Nanostring N counter analysis. The hypothesis tested is an immune predisposition to disease. The authors fail to find any supporting evidence across 20 tumours in each group for this hypothesis. The numbers included are small.

Dear reviewer, we thank you for your relevant comments. We are replying to each comment below.

The tumours are matched, but formal propensity scoring does not appear to have been done? 

We believe that proper cohort selection and matching by key factors is arguably more intuitive than propensity scoring and provides transparency. We believe the groups are representative and well matched and that little would be added by propensity scoring.

The findings are unsurprising. 

We partly agree, but if we don't do the analyses in a formal way, how will we ever know if it is as we expect? We found a void in the literature in an area we think is interesting and important.

The groups are small and not powered to observe a difference. 

This is an exploratory study where the effect size is difficult to determine in advance but based on our experience in immunohistochemistry and genetic expression analyses we have a strong belief that these homogenous and well-selected cohorts are providing enough discriminatory power.

Immunoscore or equivalent is not presented. 

The Immunoscore assessment is not open source and hard to replicate. In addition, we would like to argue that what we do is actually an improved variant of Immunoscore. We don't just give a 0, 1, or 2 score, but an exact quantification of several distinct T cell types, not just CD3 and CD8 as in the immunoscore. What we miss, however, is the distribution of T cells in the invasive margin, but on the other hand, we make a distinction between tumor cells and stroma, which is hard to capture with the Immunoscore. 

The site of infiltrate is not stated and this can influence outcome. 

Thank you for pointing this out. We have added a passage in the manuscript explaining that we are mainly exploring the tumor center, not the invasive margin. 

Overall, this paper in my opinion does not merit publication in its present form due to lack of sufficient interest. 

We acknowledge the scarcity of positive findings but refrain from preconceptions and also appreciate the importance of avoiding publication bias.

though I acknowledge the robust nature of the processes employed. 

Thank you, we appreciate this comment. 

Round 2

Reviewer 4 Report

Thank you for your responses. I appreciate your responses to my review and appreciate that the other reviewers feel that changes are not required.